# Vegetation dieback in the Mississippi River Delta triggered by acute drought and chronic relative sea-level rise

Tracy Elsey-Quirk [1] ✉, Austin Lynn[1,4], Michael Derek Jacobs[1], Rodrigo Diaz[2], James T. Cronin [3], Lixia Wang[1], Haosheng Huang [1] & Dubravko Justic[1]

Vegetation dieback and recovery may be dependent on the interplay between infrequent acute disturbances and underlying chronic stresses. Coastal wetlands are vulnerable to the chronic stress of sea-level rise, which may affect their susceptibility to acute disturbance events. Here, we show that a large-scale vegetation dieback in the Mississippi River Delta was precipitated by saltwater incursion during an extreme drought in the summer of 2012 and was most severe in areas exposed to greater flooding. Using 16 years of data (2007–2022) from a coastwide network of monitoring stations, we show that the impacts of the dieback lasted five years and that recovery was only partial in areas exposed to greater inundation. Dieback marshes experienced an increase in percent time flooded from 43% in 2007 to 75% in 2022 and a decline in vegetation cover and species richness over the same period. Thus, while drought-induced high salinities and soil saturation triggered a significant dieback event, the chronic increase in inundation is causing a longer-term decline in cover, more widespread losses, and reduced capacity to recover from acute stressors. Overall, our findings point to the importance of mitigating the underlying stresses to foster resilience to both acute and persistent causes of vegetation loss.

Vegetation diebacks have been reported in ecosystems ranging from forests, deserts, mangroves, salt marshes, seagrass meadows, and others[1–6]. Dieback events have the potential to lead to longer-term population collapse, loss of communities, and ecosystem state changes[7], although some systems can fully recover[1]. The causes of major vegetation mortality events are typically attributed to disease outbreaks, insect infestations, or acute episodic climate-mediated impacts such as severe droughts or flooding, e.g.,[8–12]. However, the propensity for underlying chronic stresses to increase ecosystem vulnerability to acute disturbances is still poorly understood[7,13,14]. Coastal wetlands are particularly vulnerable to chronic inundation stress caused by sea-level rise. Over the last century, approximately 50% of the global coastal wetland area has been lost[15] primarily due to

direct human impacts, but relative sea-level rise (RSLR) poses a compounding threat that is predicted to intensify in the future[16–18]. Though wetlands can be resilient to moderate rates of sea-level rise largely because of the vegetation, which traps sediment and contributes organic matter to the soil thus, allowing wetland elevation to equilibrate to increased flooding[19–21], rates of RSLR that exceed thresholds of plant inundation tolerance cause wetland submergence[17,22]. While wetland response to RSLR is fairly well documented, the effects of acute impacts of infrequent climate-driven events such as droughts are less known, especially in areas where high rates of sea-level rise pose a chronic and escalating stress[1].

The majority of acute wetland plant dieback events have been attributed to either excessive flooding or severe drought/low water

[1]Department of Oceanography and Coastal Sciences, Louisiana State University, Baton Rouge, LA, USA. [2]Department of Entomology, Louisiana State University Agricultural Center, Baton Rouge, LA, USA. [3]Department of Biological Sciences, Louisiana State University, Baton Rouge, LA, USA. [4]Present address: College of Agriculture and Life Sciences, Cornell University, Ithaca, NY 14850, USA. ✉e-mail: tquirk@lsu.edu

conditions[1,12,23,24], both of which can lead to the accumulation of phytotoxins in the soil[25,26]. Flooding has been implicated in the dieback of *Phragmites australis* in freshwater wetlands in Europe[12,27] and the U.S. Great Lakes[28] and *Spartina alterniflora* in a salt marsh in Texas[29], where a lack of oxygen forces anaerobic root respiration and the build-up of reduced ions that are toxic to plants. In saline environments, or areas with sufficient sulfate, anaerobic conditions foster the accumulation of sulfide, which can lead to mortality particularly when combined with high salinity that causes osmotic and ionic salt stresses. Severe drought or low water conditions can also lead to hypersaline conditions and/or potentially soil acidification through the oxidation of reduced chemical species[23,26]. Nutrient-loading, herbivory, and pathogen infection have also been implicated in wetland diebacks, but almost all in combination with atypical flooding or dessication[24,30,31]. A comprehensive review of *Spartina alterniflora* dieback sites across the U.S. Atlantic and Gulf coasts indicates that dieback throughout the southern part of the range co-occurred with severe drought[1]. Further, dieback almost always occurred in low-elevation areas with waterlogged soils in the marsh interior[1]. The concentration of drought-related dieback incidences at low marsh elevations suggests an important role of underlying flood stress yet to be quantified and incorporated into models of wetland plant dieback.

A major constraint for identifying the causes of vegetation dieback and quantifying tolerance thresholds is the lack of field monitoring data. Here, we use a valuable 16-year dataset of field measurements (2007–2022) from a network of monitoring stations across the Louisiana coast to study a recent vegetation dieback event in the Mississippi River Delta. Vegetation dieback was first reported in the late summer of 2016 affecting *Phragmites australis* (Cav.) Trin. ex Steud in the modern Plaquemines-Belize "birdsfoot" delta (BFD)[32], where the Mississippi River meets the Gulf of Mexico. However, our subsequent analysis shows that dieback began several years prior in 2012, affected multiple species, and marshes outside of the BFD. Dieback areas were initially characterized by large stands of dead stems followed by more advanced stages where only the remnant stem bases emerged from mudflats (Fig. 1). Initial observations of surviving *Phragmites* indicated an infestation of a non-native scale insect, *Nipponaclerda biwakoensis* (Kuwana), which specializes on *Phragmites* as its host plant[31]. Subsequent experimental studies show that scale infestation causes a reduction in productivity but not mortality[31]. Scale insects were first observed on *Phragmites* in 2017 and have since spread throughout Louisiana to neighboring states where dieback has not been reported.

In this study, we incorporated a rich dataset of annual vegetation cover, marsh elevation, water level, salinity, and surface elevation change time series afforded by the Louisiana Coastwide Reference Monitoring System network to (a) test empirical relationships between vegetation status and environmental conditions, principally flooding and salinity, (b) identify mechanisms likely contributing to plant mortality, and (c) quantify thresholds of tolerance and tipping points. Additionally, we tease apart the influence of short-term acute (i.e., pulse effects) and chronic (i.e., press effects) disturbances. Our analysis shows that underlying chronic flood stress predisposed low salinity marshes to an acute increase in salinity associated with low freshwater inputs and river discharge during an extreme drought. Further, by using a conceptual framework that differentiates stages and trajectories of vegetation decline

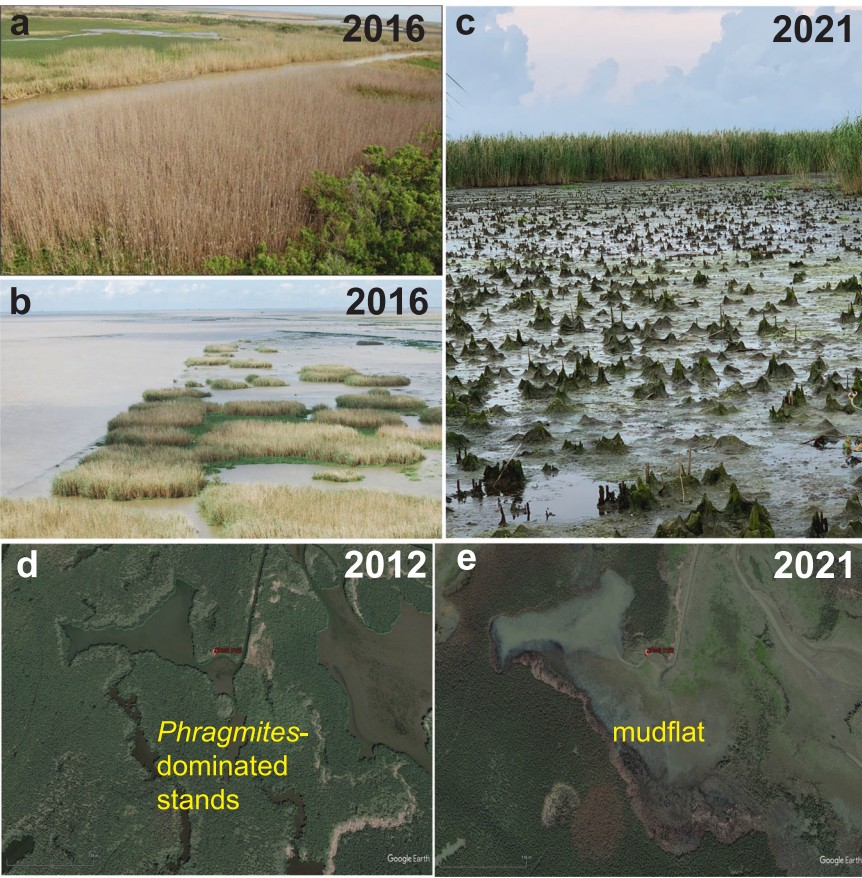

**Fig. 1 | Vegetation dieback in *Phragmites australis*-dominated stands in the BFD. a, b** Aerial images in summer 2016, **c** oblique image of a *Phragmites* stand post-dieback with a healthy stand in the background, and **d, e** satellite imagery of Coastwide Reference Monitoring System Station 0162 lower Mississippi River Delta. Photo credits: **a, b** Louisiana Department of Wildlife and Fisheries, **c** M.D. Jacobs, **d, e** Google Earth.

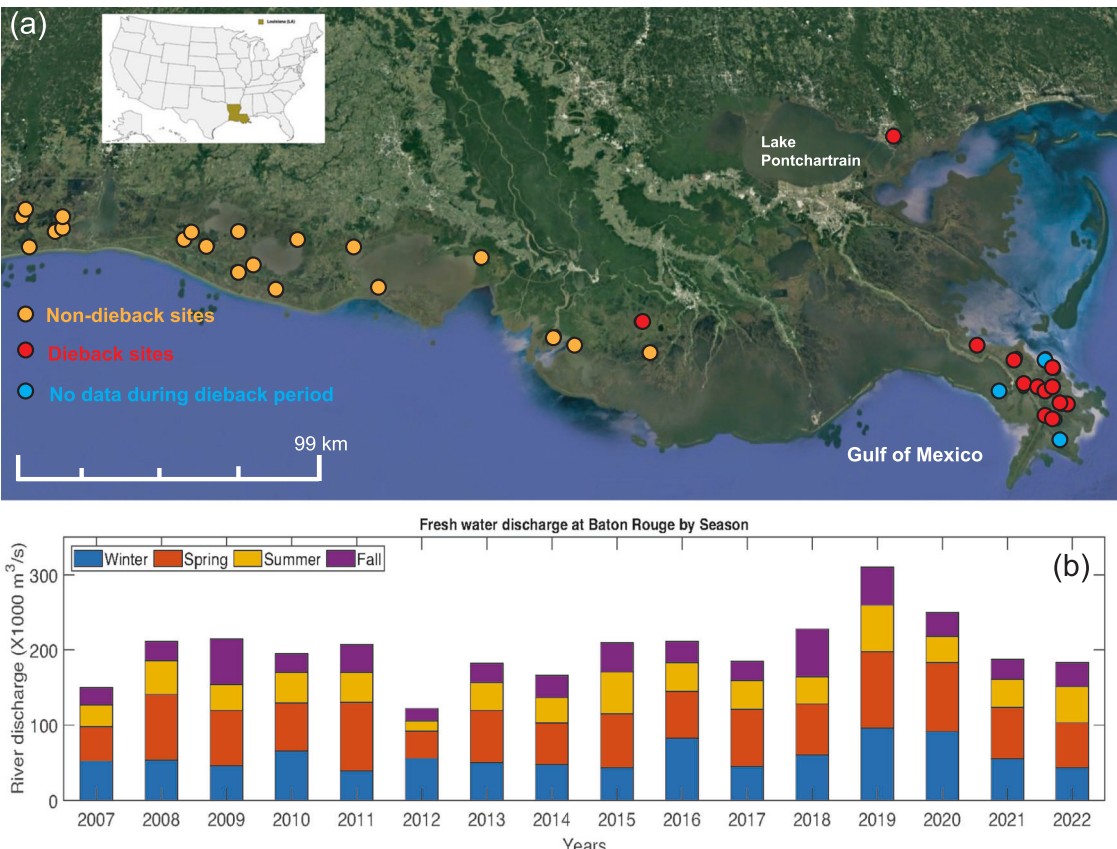

**Fig. 2 | Dieback and non-dieback marsh areas along the Louisiana coast and seasonal Mississippi River discharge, 2007–2022. a** Location of marshes that experienced dieback, no dieback, or lacked data over the dieback period, 2012–2017 based on Coastwide Reference Monitoring System data where *Phragmites australis* was a community dominant species (>20% cover), Google Earth, 2023. **b** seasonal Mississippi River discharge, Baton Rouge, LA, 2007–2022.

and recovery, we identify thresholds for dieback, conditions for recovery, and illustrate that both periodic and long-term vegetation decline is significantly related to increases in inundation. Ultimately, these findings point to specific management recommendations to reduce chronic flood stress to improve marsh resiliency to both episodic and persistent stresses.

## Study area

The Mississippi River Delta region accounts for 41% of coastal wetlands in the United States, which provides $12–47 billion in ecosystem goods and services annually[33,34]. However, these wetlands are extremely vulnerable to climate change and human impacts. A combination of rising sea levels, wave and storm-driven erosion, and rapid subsidence are exacerbated by human-built levees and dams causing a reduction in sediment supply, subsurface fluid extraction, and canal construction. All of these factors contribute to extremely high rates of coastal land loss up to 100 km² per year[35–39]. Accounting for land subsidence, recent rates of relative sea-level rise across the Mississippi River Delta average $1.3 \pm 0.9$ cm yr$^{-1}$ [40], which may exceed the estimated threshold for marsh survival (0.4–0.9 cm yr$^{-1}$)[41,42]. While over the last 7500 years, the Mississippi River has formed six major river deltas, and the current outlet of the main channel forms the BFD. This part of the coast is of high socio-economic importance as the Mississippi River serves as the main shipping route to the U.S. interior with over $150 billion per year in agricultural goods, chemicals, machinery, timber, coal, and steel transported through the main passes of the BFD[43]. The BFD is particularly vulnerable to RSLR because sediment input to delta marshes has been drastically reduced by upstream dams and channel protection measures[38]. Under conditions of reduced sediment supply and high rates of

RSLR, vegetation loss due to a disturbance or dieback event can further hasten marsh submergence through feedbacks including a reduction in sediment trapping and surface accretion, loss of root growth and turgor, decomposition of organic matter and peat collapse[44–46].

## Results

### Vegetation dieback, recovery, and longer-term decline

Vegetation cover data at Coastwide Reference Monitoring System stations were examined at multiple spatial scales: (1) in the BFD ($n = 14$); (2) at all wetlands where *Phragmites* was present ($n = 74$); (3) at sites where *Phragmites* comprised >20% cover for at least one year ($n = 33$; Fig. 2a); and (4) at a random subset of saline, brackish and freshwater marshes without *Phragmites* present ($n = 30$). Over the 16-year monitoring period, a significant dieback beginning in 2012, a year of significantly low Mississippi River discharge, was concentrated in the BFD and in two sites outside of the BFD where *Phragmites* was a dominant species, (Figs. 2, 3, Supplementary Fig. 1). Prior to the dieback, *Phragmites* comprised approximately 20–80% of the species coverage at dieback sites with a mix of other species making up the remaining percentage (Fig. 3). Coastwide Reference Monitoring System data show that the dieback began in 2012 and, in most locations, lasted five years until 2017. Dieback was characterized by an almost complete loss of *Phragmites* and a significant decline in other species (Fig. 3). Dieback sites in the BFD experienced only a partial recovery in 2017 while the two sites outside of the BFD had a full recovery to pre-dieback vegetation cover by 2017 (Fig. 3; Supplementary Fig. 1). Over the 16 years, marshes in the BFD also experienced a longer-term decline in species richness, total vegetation cover, and *Phragmites* cover concurrent with a shift to more flood-tolerant species, specifically *Colocasia esculenta*

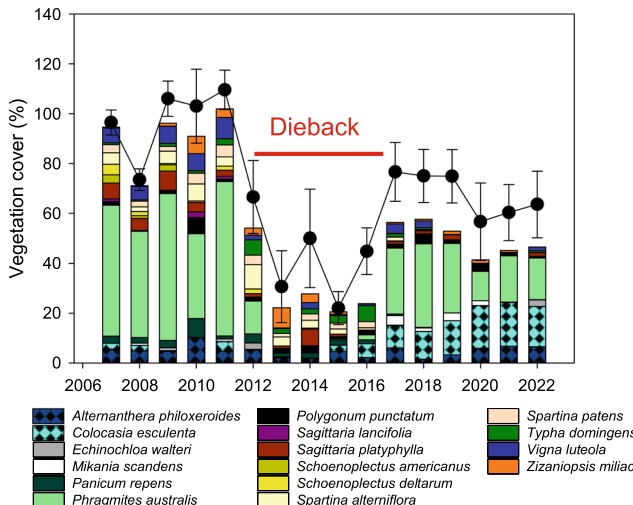

**Fig. 3 | Time series of vegetation dieback in the Mississippi River Delta.** The black scatter line represents the total annual vegetation cover (mean ± standard error) across sites ($n = 16$). The stacked bars are mean species cover data for all dominant and frequent species. Source data are provided as a Source Data file.

(Fig. 3). Stands with the highest *Phragmites* cover before the dieback in the BFD experienced a progressive decline with $79 \pm 5\%$ *Phragmites* coverage in 2011, $47 \pm 9\%$ in 2017 and $25 \pm 3\%$ in 2022, representing an average decline of 68% over the 16-year monitoring period (Figs. 1d, e and 3; Supplementary Fig. 1).

## Dieback sites experience greater flooding and lower salinities than non-dieback sites

All dieback stations were dominated by *Phragmites* yet twenty stations dominated by *Phragmites* located farther west along the Louisiana coast did not experience dieback (Fig. 2a; Supplementary Fig. 1). To investigate whether flooding differed among dieback and non-dieback sites, we used daily marsh inundation, a measure of water level relative to the marsh surface, using a combination of water level (referenced to NAVD88) and marsh elevation (referenced to NAVD88) at each Coastwide Reference Monitoring System station (see "Methods" section for details). We tested the strength of relationships between inundation, salinity, and vegetation during different time periods over the growing season and found that flooding and salinity dynamics from May through July had the most profound influence on summer vegetation cover, as compared to later in the growing season (August through October) or averaged over the entire growing season (March through October), likely due to this being a critical period for plant growth.

A comparison of marsh inundation and salinity May through July from 2008 through 2021 between dieback and non-dieback sites revealed that (1) dieback sites are persistently subject to greater flood depths and lower salinities than non-dieback areas; (2) dieback areas had exceptionally high mean daily and maximum salinity in 2012; (3) marsh inundation is increasing over time at all locations; and (4) tolerance to high salinity is greater under less flooded conditions as reflected by the high vegetation coverage and low dieback in saltier but less flooded sites (i.e., non-dieback sites; Fig. 4). Across the time series, marsh inundation depth averaged 15 cm higher in the dieback areas than non-dieback areas (Fig. 4a). Maximum flood depths May through July, which represent occurrences of storm flooding and, in the BFD, river discharge, were similar between dieback and non-dieback areas indicating that differences are due to average conditions, not extreme events (Fig. 4c). Salinities are typically very low, less than 1 psu in dieback marshes because of the proximity of most sites to the Mississippi River, but average between 3 and 5 psu and up to 10 psu

in the non-dieback areas (Fig. 4b, d). In 2012, mean and maximum salinity in the dieback areas were exceptionally high, averaging 4.5 and 13 psu, respectively May–July.

Though dieback marshes have experienced greater flood depths over the last 16 years, marsh inundation has increased at all sites (Fig. 4a). As marsh flooding increased in non-dieback areas, salinities have declined suggesting greater freshwater inputs (Fig. 4a, b). However, with an increase in inundation across the region, the vulnerability to acute disturbances such as salinity pulses from droughts or storms is amplified both spatially and temporally.

## Acute dieback caused by drought-induced salinity incursion and exacerbated by tropical cyclones

Vegetation dieback coincided with the second most severe U.S. drought on record[47,48] and Tropical Storm Debby in late June 2012[49] causing elevated salinities throughout the early part of the growing season, May–July (Figs. 3, 4, and 5a). The drought resulted in low Mississippi River discharge and prolonged high salinities during the spring and summer. Tropical Storm Debby affected the dieback region for several days beginning approximately June 20, 2012. Our analysis shows that high background salinities associated with the drought likely facilitated the unusually high salinities during this early-season tropical storm. Except for Tropical Storm Debby, all early tropical cyclones in June and July across the 16-year time series were freshwater flooding events (Supplementary Fig. 4). By parsing out the drought- and storm-related flooding and salinities (see "Methods" section), we find that drought-related salinities persisted for an extended period May through July with over 20 days of >5 psu and over 10 days with salinity exceeding 10 psu (Supplementary Fig. 5). The close proximity of the Loop Current to the mouth of the Mississippi River in July also brought high salinity water near the coast (Supplementary Fig. 6). Marsh inundation in May 2012 was lower than any other year from 2008 to 2022 ($p < 0.01$), when the Mississippi River discharge was very low (Fig. 2b) and 24% of the contiguous U.S. was experiencing severe to extreme drought[47]. However, despite the relatively low water, dieback marshes averaged 13 cm greater water depths than non-dieback marshes and remained saturated with high salinity water (Fig. 4a, b). Tropical Storm Debby exacerbated these conditions by causing 45 cm of marsh inundation and 17 psu salinity across monitoring locations for several days (Fig. 5b, Supplementary Figs. 3 and 4). Hurricane Isaac followed the initial detection of the dieback in late August and caused saltwater flooding of almost 1 m across the dieback marshes with maximum salinities averaging 25 psu, greater than any other tropical cyclone in the 16-year time series (Fig. 5b, Supplementary Fig. 4). The high flooding and unprecedented salinities during Hurricane Isaac likely contributed to the protracted 5-year impact on the vegetation.

## Thresholds for dieback and conditions related to decline and recovery

Determining empirical relationships between vegetation status and environmental conditions can be complicated by hysteresis in vegetation responses to ephemeral disturbances. For example, vegetation loss can be prolonged even after the stressor is removed and conditions have been restored due to biological lags in regeneration capacity, recolonization, and growth[50]. Because vegetation response may be delayed or on a trajectory of decline or recovery trailing or disparate from the immediate environmental conditions, plant status may not directly reflect abiotic conditions at a particular time. Here, we used a conceptual framework that classifies distinct vegetation states based on relative percent cover over time and trajectories from previous years modified from a model describing hurricane impacts on coral reef declines[51]. This framework was used to compare environmental conditions among states (Peak, Decline, Dieback, Impact, Recovery, and Resumption) and test relationships between vegetation

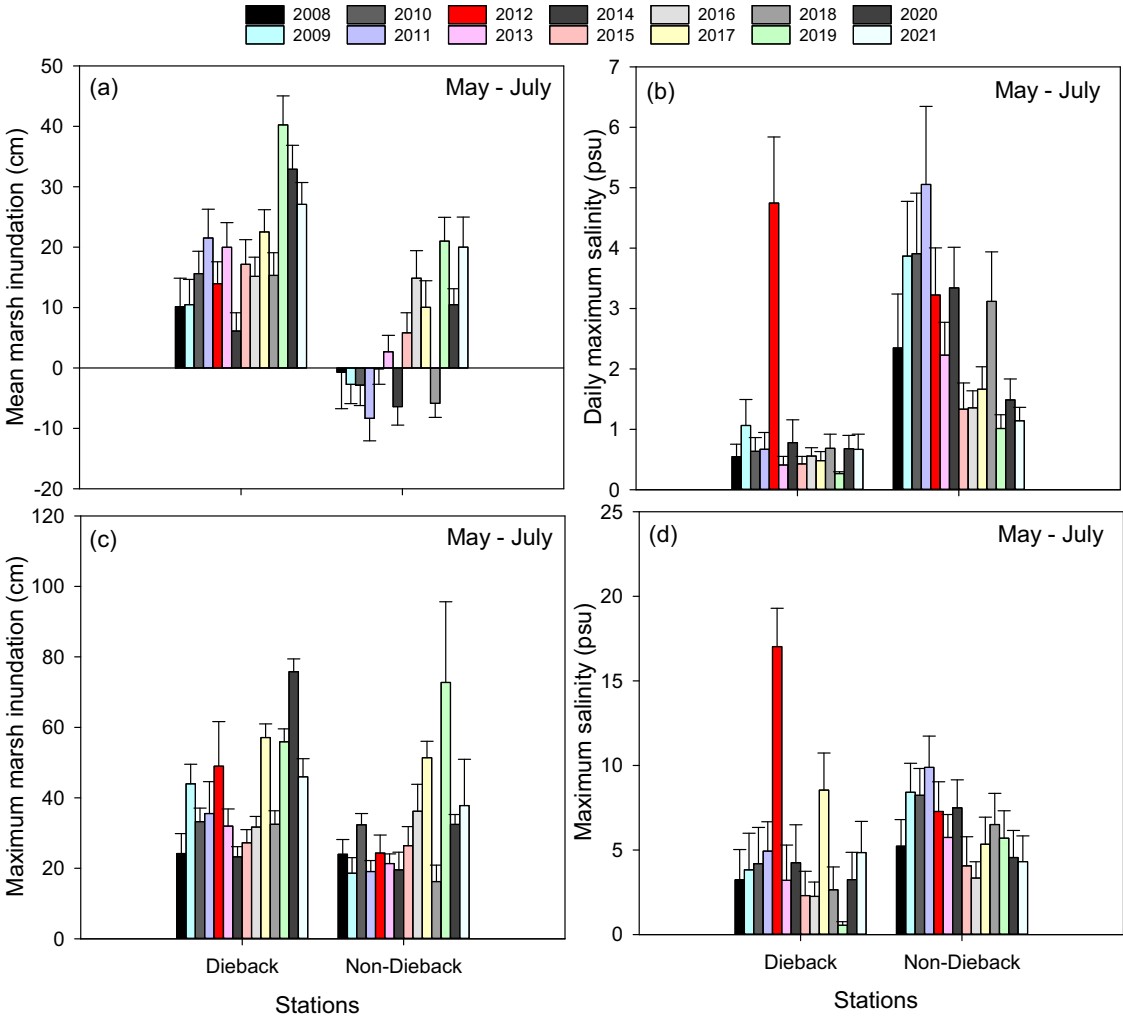

**Fig. 4 | Annual marsh inundation and salinity May through July in dieback and non-dieback Coastwide Reference Monitoring System stations where *Phragmites australis* was a significant part of the vegetation community. a** Mean marsh inundation, **b** mean daily maximum salinity, **c** mean maximum marsh inundation, and **d** mean maximum salinity May through July 2008–2021 across dieback ($n = 16$) and non-dieback ($n = 20$) sites. Dieback occurred in marshes where *Phragmites* comprised >20% of the vegetation community for at least one year of monitoring therefore this comparison was focused on marshes composed of >20% *Phragmites* for at least one year. Values are means ± SE.

cover and environmental variables within states (Fig. 6a, b). For each Coastwide Reference Monitoring System station where dieback occurred, total vegetation cover each year was classified as one of six vegetation states. The Peak state occurred during years with the highest (90th percentile) vegetation coverage for each site. The Decline state was characterized by years with a reduction in cover which was lower in magnitude than the Dieback state. Vegetation cover during Declines averaged 78 ± 8%, (a 28% decline from Peak coverage; $p < 0.01$), and 51 ± 13% during the Dieback. *Phragmites* cover ranged from 4–60% during Declines and 0–3% during the Dieback. Distinguishing lower magnitude vegetation declines from the dieback allows us to examine the potential for declines and dieback to be affected by different drivers. An Impact state was characterized by years with low vegetation cover following a Decline or Dieback when abiotic conditions (i.e., inundation and salinity) were restored. The Recovery state occurred in years with an increase in cover following Declines, Diebacks, or Impact periods, and Resumption represented a stasis following a Recovery that did not reach Peak cover (Fig. 6a). It is important to acknowledge that our system is also experiencing a declining baseline (Figs. 3, 6b) but for purposes of this analyses, we included only conditions during the highest peak coverages using the stable population framework (Fig. 6a).

Peak vegetation cover averaged 109 ± 4% (Fig. 3; Fig. 7; $p < 0.01$). Vegetation cover measured at Coastwide Reference Monitoring System stations are the sum of individual species covers that include under-, mid-, and overstory canopy and vine species and therefore can exceed 100%. Peak cover occurred when the marsh surface was flooded approximately 50% of the time over the growing season and salinity averaged less than 2 psu and was above 10 psu for less than 10 days (Fig. 6c, d; Supplementary Fig. 7). The Dieback was characterized by a high magnitude of vegetation loss and, for the dominant species, *Phragmites*, 80 to 100% lower coverage (Figs. 3, 6). Marsh inundation depth and percent time flooded was similar during Peak and Dieback states and higher during the Impact, Decline, Recovery and Resumption periods (Fig. 6c, d; Supplementary Fig. 7). During the dieback, average salinity over the growing season was over four times greater than during other states ($F_{5, 173} = 120.5$, $p < 0.0001$) and salinities above 10 psu lasted for an average of 27 days as compared to 5 days during Peak coverage (Fig. 6d; Supplementary Fig. 7b). Lesser magnitude Declines occurred both before and after the high magnitude Dieback primarily in 2008, 2010, and 2020 and over the long-term (Fig. 3). Impacts rarely occurred following a decline but were consistent across sites following the Dieback. Vegetation cover during the Impact state was statistically similar to the Dieback and for *Phragmites* averaged

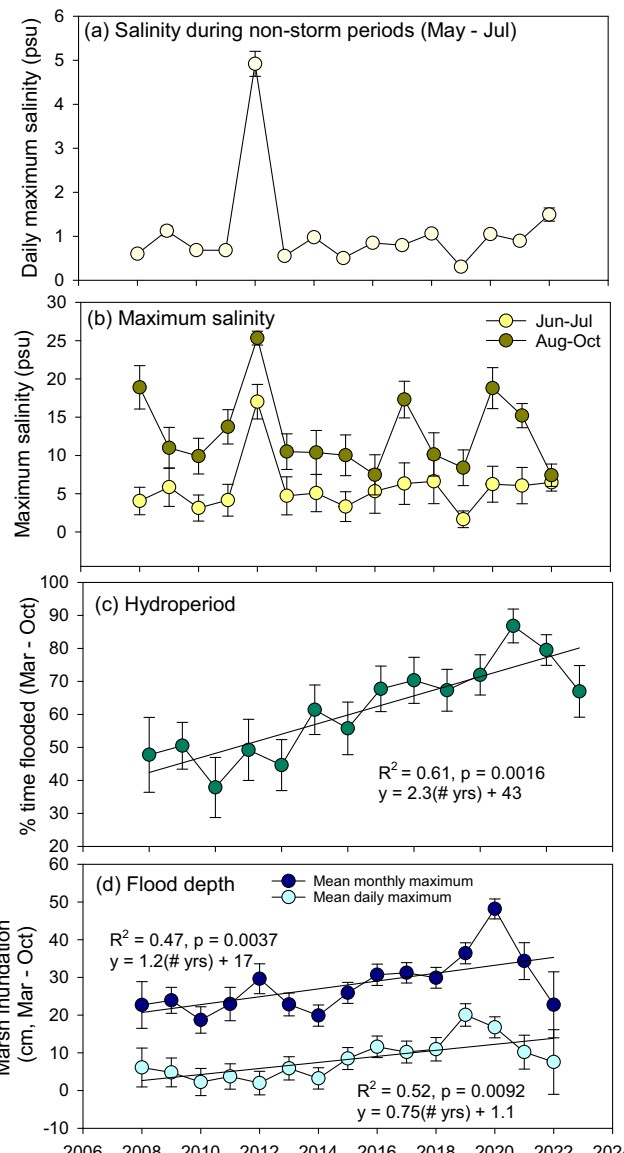

**Fig. 5 | Time series of salinity and hydrological conditions across marsh dieback areas in the Mississippi River Delta. a** Mean daily salinity during non-storm periods, May–July, **b** mean maximum salinity during Jun–July and August–October periods, **c** generalized linear model of mean percentage of time the marsh is flooded over the growing season and **d** generalized linear models of mean daily and monthly maximum marsh inundation depth over the growing season across 14 Coastwide Reference Monitoring System stations in the BFD (*n* = 14, ±standard errors). Source data are provided as a Source Data file.

$1.6 \pm 0.6\%$. Recovery states occurred following Decline, Dieback, and Impact states notably in 2009, 2011, 2017, and 2021. Resumptions occurred in 2018, 2019, and 2022. Using these population states, we examine relationships and threshold dynamics across and within states (Fig. 7).

**Thresholds and relationships**
Focusing on *Phragmites* cover dynamics and tolerances, we found no threshold marsh inundation level that characterized the transition between peak cover and dieback, which occurred during a period of relatively low water ($p > 0.05$). The transition from peak to dieback cover was described by a significant sigmoidal relationship with mean daily maximum salinities over the growing season (sigmoidal model: $y = a/(1 + e^{(-(x - c)/b)})$; where $a = 147$, $b = -1.78$, and $c = 4.91$; Adj

$R^2 = 0.54$, $p = 0.0003$)) indicating a threshold value of 7.2 psu (Fig. 7a). This salinity threshold represents an average condition over the growing season when dieback was initiated but not an absolute threshold for mortality.

The magnitude of *Phragmites* cover loss during interannual Declines (primarily in 2008, 2010 and 2020), which was less than that during the dieback, was best explained by maximum marsh inundation May through July indicating that early-season extreme flooding from high river discharge and/or tropical cyclones reduces plant density during that summer (Fig. 7b). The most severe and spatially consistent decline in cover occurred between summer 2019 and 2020 following excessive freshwater flooding from Tropical Storm Cristobal in June 2020 and five succeeding hurricanes, which caused elevated water levels and salinities July through October, and likely contributed to a limited recovery of *Phragmites* (<10%) by summer of 2021 (Supplementary Fig. 1). Overall, maximum marsh inundation May–July explained approximately 60% of the variation in vegetation declines (Fig. 7b). And while declines in cover were also statistically related to mean inundation averaged across the growing season, the relationship was strongest with maximum inundation May through July. *Phragmites* recovery from dieback and declines occurred when salinities were low with an optimal flood depth between 20 and 30 cm from May through July based on a parabolic relationship ($R^2 = 0.19$, $p < 0.05$) indicating that relatively moderate early-season flooding increased the percent of recovery (Fig. 7c).

**Longer-term vegetation loss related to an increase in inundation**
Local RSLR across the BFD, calculated using water level time series data, has been increasing at a rate of $2.7 \pm 0.1$ cm yr$^{-1}$ over the last 16 years ($R^2 = 0.66$, $P < 0.01$, $y = 2.66x + 0.04$;) due to a combination of subsidence and eustatic sea-level rise. However, marsh inundation depends on marsh surface elevation and the rate of marsh elevation change. Marsh elevation change rates in the BFD, measured biannually using surface elevation tables, range from 0.3 to 5.8 cm yr$^{-1}$ and average $2.1 \pm 0.4$ cm yr$^{-1}$. Marsh inundation, which incorporates water level, surface elevation change, and surveyed marsh elevations, across the delta during the growing season has been increasing at a rate of 0.75 cm yr$^{-1}$ since 2008 (Fig. 5d). Monthly maximum flood depth has also increased but at a much higher rate of 1.2 cm yr$^{-1}$. Further, the percentage of time that the marshes are flooded (i.e., the percentage of days that the mean water level was above the marsh surface), has increased from an average of 43% to 75% (Fig. 5c).

The temporal trends in vegetation decline over the 16-year time period correlate to increases in marsh inundation and hydroperiod (ANCOVA: $R^2 = 0.34$, $p < 0.0001$; Figs. 3, and 5c, d) and further, total vegetation cover declines significantly with an increase in marsh inundation depth over the growing season (Fig. 8). Excluding the Dieback state when water levels were markedly low, vegetation cover declined with an increase in inundation across all population states indicating that the degree of Decline, Recovery, and Resumption and absolute extent of vegetation cover are governed largely by flooding. The long-term trends of increasing marsh inundation in the BFD coincide with shifts in the vegetation community to fewer, more flood-tolerant species (Fig. 3). Species richness averaged 11–14 species before 2012 and has ≤7 species over from 2019–2022. The decline in species richness is accompanied by a reduction in robust emergent species such as *Phragmites*, *Sagittaria platyphylla*, *Spartina alterniflora*, and *Spartina patens* and an increase in the flood-tolerant *Colocasia esculenta* (Fig. 3).

**Discussion**
The Mississippi River Delta region is one of many coastal areas around the world vulnerable to wetland loss due a combination of human management and climate-driven changes[34–37]. Vegetation dieback in this system was associated with an acute increase in salinity

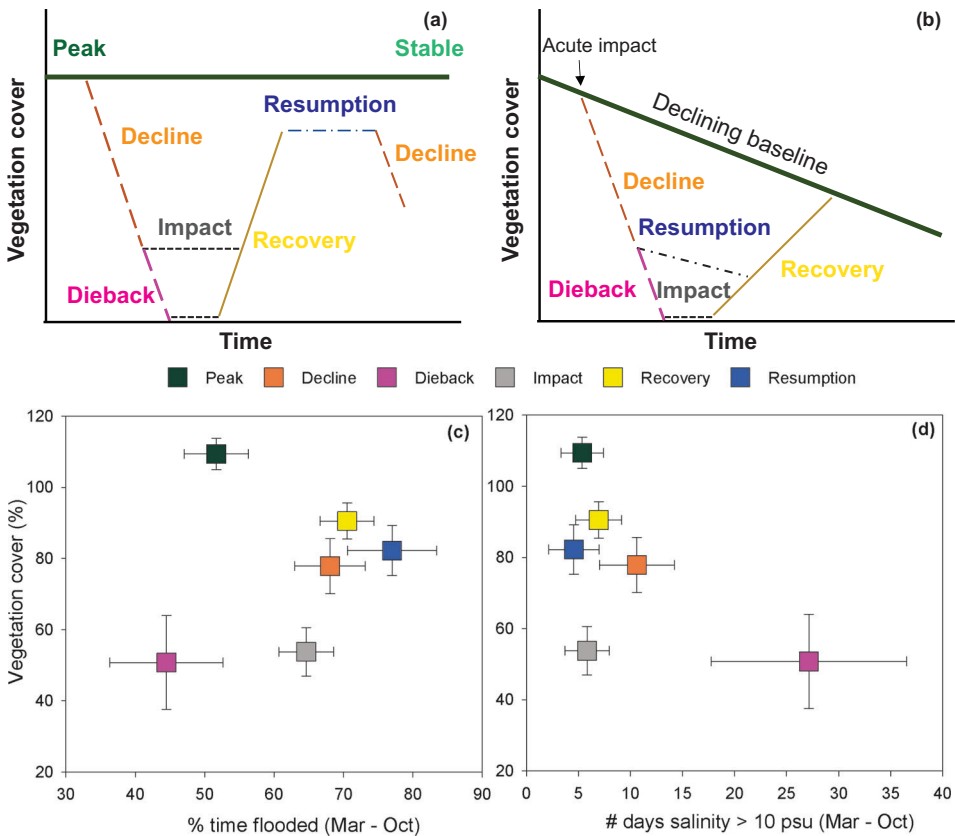

**Fig. 6 | Vegetation states and impact of marsh inundation and salinity.**
**a** Conceptual framework of vegetation states based on trajectories of vegetation cover with a stable steady-state baseline modified from ref. 51, **b** conceptual framework of vegetation states based on trajectories of vegetation cover with a declining baseline modified from ref. 51, **c** relationship between vegetation cover and hydroperiod during the growing season for different vegetation states, **d** relationship between vegetation cover and duration of salinity >10 psu during the growing season for different vegetation states. Data are means ± standard error.

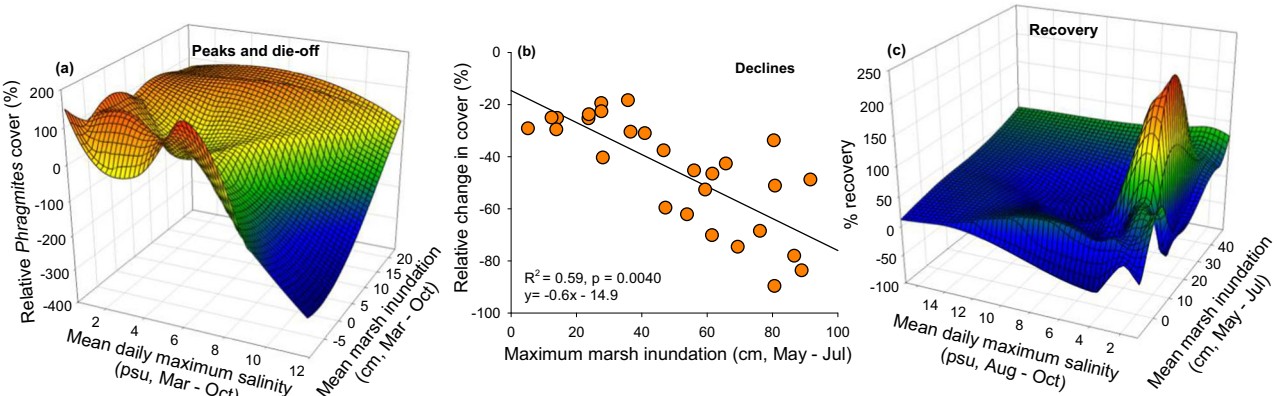

**Fig. 7 | Relationships between *Phragmites* cover dynamics and marsh inundation and salinity. a** 3-D mesh interpolation of the combined effects of marsh inundation and maximum daily salinity over the growing season on *Phragmites* dieback (includes sites with initial cover above 50%), **b** relationship between maximum marsh inundation (May–July) and change in percent cover during declines based on simple linear regression **c** 3-D mesh interpolation of the combined effects of marsh inundation (May–July) and daily maximum salinity (August–October) on *Phragmites* recovery. Note y-axis is reversed to show the low cover at high salinity and low inundation.

particularly during the early to peak growing season. Here we show that recovery from acute causes of vegetation loss can occur but is tempered by underlying chronic stresses to the system. Both short- and long-term declines in vegetation cover and richness were associated with high levels of marsh inundation because of marsh subsidence and sea-level rise. Therefore, managing the source(s) of chronic stress, in this case, RSLR, is imperative to facilitate resilience to

both short-term disturbances as well as the underlying causes of longer-term decline.

Though the focus of this study is on the impacts of salinity and inundation on vegetation loss, there are certainly other anthropogenic changes occurring in the Mississippi River Delta that are likely important stressors and threats to plants. Nutrient inputs from agricultural, sewage and urban-development runoff leading to

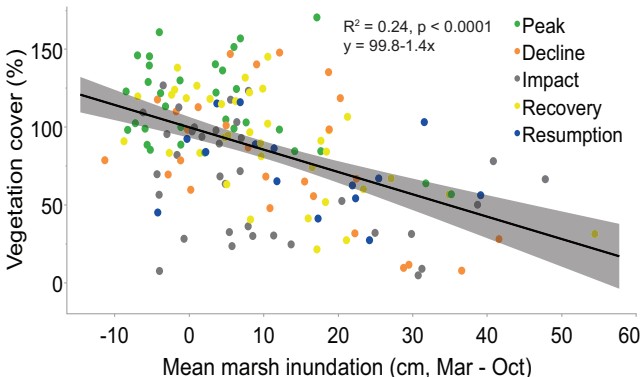

**Fig. 8 | Relationship between total vegetation cover and mean marsh inundation during the growing season at dieback Coastwide Reference Monitoring System stations in the Mississippi River Delta.** Generalized linear model based on linear regression of mean marsh inundation and vegetation cover. The shaded area represents 95% confidence interval from the mean. Source data are provided as a Source Data file.

eutrophication[52–54], toxic heavy metals such as copper, iron, and zinc[55], frequent oil spills[56], invasive plant species (e.g., *C. esculenta*[57,58]) and rising temperatures[59] are all potential stressors to plant communities in coastal marshes. In 2010, two years prior to the documented dieback, the BFD and other areas along the northern Gulf coast were impacted by the BP-Deepwater Horizon oil spill. In marshes exposed to oiling, vegetation mortality was concentrated at the edge where oil coverage was greatest and, in some locations, recovery of marsh grasses in previously oiled areas occurred within 18 months[60]. An analysis of the BFD marshes showed little impact of oil on the vegetation attributed to relatively low oil residence time due to river discharge and limited oil deposition on the soil[61]. Though there is the potential for the oil spill to have impacted dieback areas two years before the dieback, the strong temporal correlations shown here suggest that salt intrusion and underlying flood stress were important. Nutrient concentrations, particularly nitrate concentrations, have increased in the Mississippi River over time[62] and high nutrient availability has been hypothesized to contribute to *Phragmites* dieback in other systems[12,27]. Nutrients cause an increase in organic matter production which, under anaerobic conditions, can intensify the production of organic acids and sulfides[27] and while not directly tested here may also contribute to vegetation stress in the BFD. While we show that recovery of vegetation is influenced by RSLR, it is not the only chronic stressor likely to slow recovery. Introduction of the Roseau cane scale (*N. biwakoensis*) into the Mississippi River Delta, a specialist of *Phragmites* and first reported in 2017[32], is clearly slowing recovery of dieback sites. In a common-garden experiment, low to moderate scale densities (≈20 per m of the stem) reduced end-of-season biomass by 28% relative to plants with few scales (4 per m of stem). Peak densities of scales, occurring in late summer, have consistently exceeded 150 per m of stem since 2017[32]. Scale densities among *Phragmites* stands in the Mississippi River Delta are negatively correlated with the normalized difference vegetation index (NDVI); a measure of standing plant biomass (I. A. Knight, G. Suir and R. Diaz, unpublished data). All of these stressors have the potential to interact, elevating their chronic effects to an acute level.

Greater marsh inundation affected the susceptibility to vegetation dieback and while not the cause of acute dieback, caused interannual and longer-term population and diversity declines. Vegetation dieback beginning in 2012 was concentrated in the BFD where both rates of RSLR and marsh elevation change are relatively high[40]. But, as compared to non-dieback sites farther west, the BFD marshes are lower in elevation and affected by both river and Gulf of Mexico hydrodynamics (Supplementary Fig. 8). High flood depths averaged across

the growing season were strongly related to low vegetation cover, while the magnitude of annual declines in cover was influenced by early-season high maximum marsh inundation. Elevated water levels were attributed to dieback of *Phragmites* in Europe[12,63], several native marsh plant species in the southeastern United States[56,64], and mangroves on the Indo-Malayan coast[65]. Our data show that high river discharge and marsh inundation in the BFD induces an immediate short-term loss of vegetation, yet the chronic increase in average marsh flooding is contributing to a slower longer-term decline. Recovery of *Phragmites* following declines and dieback was dependent on an optimal moderate level of spring flooding and in restoration trials, *Phragmites* plantings in dieback areas had low survivorship in areas of greater water depth[66]. This relationship with flooding is similar to responses of other wetland plants to experimental sea-level rise[19] and supports findings in salt marshes where vegetation recovery following a disturbance slows with an increase in seawater inundation[67].

Acute dieback occurred during an extreme drought with the proximate cause of saltwater intrusion. Most of the dieback sites were located adjacent to the Mississippi River where low discharge caused upriver transport of the salt wedge. While this dieback was associated with an extreme drought, other river management activities that lower freshwater discharge (e.g., upstream dams and reservoirs, diversion of river water) can also cause low water conditions in the river. To accommodate larger ships, the U.S. Army Corps of Engineers has deepened the Mississippi River Ship Channel to a minimum of 53 feet. Deep navigation channels can facilitate the upstream movement of saline water particularly during low river discharge.

The dieback targeted *Phragmites*-dominated marshes. Marshes near dieback sites dominated by other species including more salt-tolerant species, were relatively stable during the dieback period. Coastal plant diebacks tend to impact single species. Examples include diebacks of *Spartina alterniflora*, *Phragmites australis*, and the seagrass *Thalassia testudinum* that all typically grow in large monocultures[1,6,12]. Seagrass dieback in Florida Bay, for example, targeted dense stands of *Thalasia*, while sympatric species remained unaffected. Turtlegrass was susceptible to dieback because of its growth habit of forming dense meadows where high biomass and respiratory demand for oxygen led to phytotoxic sulfide incursion[68]. Although it is not entirely clear why *Phragmites*-dominated marshes were exposed to dieback in the Mississippi River Delta, a similar dynamic to *Thalassia* may be occurring and previous studies have implicated the accumulation of organic acids and sulfides as the products of anaerobic decomposition in the rhizosphere[12,27,69]. This combined with high salinities, which can further facilitate the generation of sulfides, can be lethal. Salinities that exceed 15 psu in the root zone have been associated with *Phragmites* dieback prior to the onset of flowering across 27 habitats along the eastern and western coasts of Denmark[70]. We found a dieback threshold of 7 psu averaged across the growing season that represents significant time periods of relatively high salinity (e.g., 20 days ≥ 5 psu, 10 days ≥ 10 psu) during the drought and extreme salinity and flooding pulses during subsequent tropical cyclones. However, relatively high salinities and low water levels in non-dieback marshes in the present study provide further support that *Phragmites* can tolerate high salinities as long as the sediments are relatively oxidized[71].

## Implications
As we write this paper, drought conditions have caused two consecutive years of historically low water levels in the Mississippi River. Based on historic temperature and precipitation trends in the Mississippi River watershed, extreme summer droughts are predicted to become more frequent in the future in response to climate warming[72]. The loss and potentially limited recovery of marsh vegetation from a prolonged salinity intrusion event and chronic flood stress, as we have documented, can have a number of ecosystem-level consequences

**Table 1 | Methodology for data collected at Coastwide Reference Monitoring System stations used in this study**

| Data Type | Parameter | Method | Scale | Frequency |
|---|---|---|---|---|
| Vegetation | Percent cover | Braun Blanquet % cover | (10) 2 m x 2 m plots per site | Annually during peak biomass |
| Marsh elevation | Elevation relative to NAVD88 | Real-time Kinematic Global Position System Survey | At RSET; 1 per site | 2014 |
| Marsh elevation change | Surface elevation change | Rod Surface Elevation Table (RSET) | 1 RSET per site; 4 directions per RSET | 2 ×'s per year 2006–2020; 1 ×/yr after 2020 |
| Hydrology | Surface water level (NAVD88) | Submersible data logger | Channel or bay water within 200 m of marsh plots | Hourly |
| Salinity | Surface water salinity | Submersible data logger | Channel or bay water within 200 m of marsh plots | Hourly |

including the loss of breeding and foraging habitat for birds[73,74], muskrats, and the American alligator[75], changes in soil biogeochemistry[27,76,77], lower accretion rates and soil strength[78] and thus a reduced ability to keep pace with sea-level rise[21]. Vegetation plays a pivotal role in wetlands by stabilizing channel banks, facilitating sediment deposition, contributing organic matter to soil and oxygenating the rhizosphere. The consequences of mass mortality may lead to peat collapse and conversion to mudflat through the collapse of air-filled rhizomes and decomposition of organic matter. Lower elevation, more flooding, and anoxic soil conditions may limit the recovery of marsh vegetation *resulting in a state change from emergent marsh to mudflat/submerged habitat without a large input of allochthonous sediment*[45] *(*Supplementary Figs. 9–13*)*. The Mississippi River Delta is suffering from catastrophic rates of wetland loss and marshes of the BFD are disappearing rapidly. Our findings suggest that a reduction in chronic inundation through the delivery of sediment (e.g., enhanced flow through channels and connectivity to marshes, thin-layer placement) to increase marsh elevations will reduce the vulnerability of marsh vegetation to episodic disturbances and enhance recovery following disturbances.

## Methods

The dieback was initially reported as affecting *Phragmites australis* in the birdsfoot delta (BFD). To examine the scale and magnitude of the vegetation dieback, we examined the time series of species percent cover data collected annually during peak growing season since 2006 or later at Louisiana Coastwide Reference Monitoring System Stations. The Coastwide Reference Monitoring System network is comprised of 391 monitoring stations across swamp, freshwater, intermediate, brackish, and saline wetlands with data collection beginning in 2006 and continuing to the present day. To test whether the dieback is reflected in the annual vegetation cover data at Coastwide Reference Monitoring System stations, we examined the 16-year time series of species cover data at Coastwide Reference Monitoring System stations in the BFD ($n = 14$), in all marshes where *Phragmites* was present ($n = 74$), and at a random subset of saline, brackish and freshwater marshes without *Phragmites* present ($n = 30$). Dieback was detected at sites with a relatively high coverage of *Phragmites* (>20% for at least one year since 2007) and was concentrated in the BFD. Two sites outside of the BFD where *Phragmites* was dominant also experienced dieback (Fig. 2b). Summary annual data used for this study are provided as Supplementary Data 1.

Subsequent analyses were conducted to examine (1) differences between dieback and non-dieback marshes in hydrology (i.e., marsh inundation and percent time flooded – see below for detail) and salinity over time; (2) annual differences in hydrology and salinity in dieback marshes; and (3) relationships between vegetation cover and hydrology and salinity during peak, dieback, declines, recoveries, and other population states and across the time series to determine potential thresholds of tolerance. For these analyses, we used Coastwide Reference Monitoring System annual vegetation percent cover, marsh elevation (NAVD88), marsh surface elevation change, hourly

water level, and hourly salinity data. All raw data used for this study are publicly available[79].

Each Coastwide Reference Monitoring System station is comprised of a 200-m × 200-m area where marsh elevation, marsh surface elevation change are measured, vegetation plots are located and soil cores are collected and other station measures (e.g., periodic plant biomass) are taken. At each station, continuous water level, salinity, and temperature data are recorded hourly from a nearby channel or bay. Detailed information on the methodology of Coastwide Reference Monitoring System station establishment and data collection can be found in ref. 80. Table 1 below shows a brief outline of methods for vegetation and hydrology used in this study.

### Vegetation cover

Vegetation measurements at Coastwide Reference Monitoring System stations are collected during the summer, approximately June through August, each year. At each station, percent cover data is collected within ten permanent 2 m × 2 m plots along a 283 m transect using the Braun-Blanket cover class method: 0%, <1% 1–5%, 5–25%, 26–50%, 51–75%, 76–100%[81]. The site mean of the mid-point of each cover class for each species is reported for each year. Total vegetation cover of each station and cover of each layer (e.g., canopy and understory) is estimated between 0 and 100 percent. Therefore, the sum of the species cover (i.e., total cover) may exceed 100 percent because of overlapping canopies.

Assessment of percent cover data by basin illustrated that acute dieback began in 2012 and lasted until 2017 and was concentrated in the BFD ($n = 14$). Unfortunately, percent cover data was not recorded for some of the BFD Coastwide Reference Monitoring System stations during the dieback period (2012–2017). Field notes indicate dense *Phragmites* stands at the marsh edge and that the coverage in the interior plots was assumed to be 100%. However, vegetation condition at the marsh edge does not necessarily reflect the condition of vegetation in the marsh interior. All other stations in the BFD showed a significant decline in cover that lasted approximately 5 years. All but one site in the lower delta has experienced a significant longer-term decline in coverage over the 16 years.

**Phragmites haplotypes.** The vegetation on the outer fringe of the BFD consists of virtually monospecific stands of 4–5-m-tall *Phragmites*[82] with a diversity of brackish to freshwater marsh plants also present[83]. In 1968, *Phragmites* comprised 11, 57, and 39% of the salt, brackish, and fresh marsh vegetation respectively[84], and since then have been considered the most common dominant marsh species in the BFD[33]. Four distinct haplotypes of *Phragmites australis* occur along the Gulf coast from Florida to Texas[85]. Land-type (I2), occurring inland primarily along roadside ditches; Delta-type (M1), the most widespread haplotype in Louisiana marshes, EU-type (M), the notorious invasive haplotype in many areas around the world, but only found in localized patches in marshes of Louisiana, and the Greeny-type (AI) – uncommon and in localized patches. Coastwide Reference Monitoring System vegetation data does not distinguish *Phragmites* by haplotype, and

while both EU- and Delta-types are present in marshes of the BFD, the Delta-type is dominant and more widespread than the EU-type.

## Hydrology and salinity

**Raw data.** Water level and salinity datasets for this study cover a time period from approximately July 2007 through December 2022. Because only the latter part of 2007 data were available, 2008 through 2022 were used for analyses of relationships between hydrology, salinity, and *Phragmites* cover. Water level are referenced to the NAVD88 datum. Water level and salinity data are collected hourly, and for this analysis mean daily water level (i.e., average of the 24 hourly water level values) and maximum daily salinity (i.e., maximum of the 24 hourly salinity observations) were used.

**Marsh elevations.** Station elevations were surveyed in 2014 using an RTK-GPS at the RSET, the sonde and staff gauge, and the marsh surface. The elevations were relative to NAVD88 using Geoid 12A.

**Marsh elevation change.** Rod Surface Elevation Tables (RSET) were established at each Coastwide Reference Monitoring System station for measuring marsh elevation change over time. The RSET methodology[86] consists of stainless steel rods driven vertically into wetland sediments to the point of refusal and secured in concrete bound with a PVC-pipe collar. During each site visit (1–2 times yr⁻¹), an RSET arm is attached to the rod, and in each of four directions, nine pins are lowered to the wetland surface, for a total of 36 pin height measurements per site visit. Marsh elevation change is calculated by taking the mean difference between subsequent pin heights and initial pin heights across the 36 measurements. An annual rate of change is calculated for each station using a linear trend in elevation change.

**Marsh inundation.** For each Coastwide Reference Monitoring System station, we calculated "daily marsh inundation" to represent water level relative to the marsh surface using the daily mean water level (NAVD88), initial marsh elevation in 2014 (NAVD88), and marsh surface elevation change over the 16-year period. Surface elevation change is measured biannually using RSETs[80]. Incorporating marsh elevation change is essential for calculating marsh inundation in areas experiencing shallow subsidence and accretion that result in marsh elevation change over the period of study. Linear interpolation was used to estimate daily marsh elevations between elevation change measurements. Daily marsh inundation was calculated over the 15-year time series by subtracting daily marsh elevation from the mean daily water level for each station. Positive values indicate water level is higher than the marsh elevation and, thus, the marsh surface is submerged.

**Surface water salinity.** Daily maximum salinity is defined as the maximum value of the 24 hourly observations in a day, which was typically within 2 psu of daily mean salinity, was used for all analyses as maximum salinities were considered most relevant for assessing contributions to plant stress and loss of cover.

Daily mean inundation and daily maximum salinity values were averaged across Coastwide Reference Monitoring System stations over specific time periods (i.e., growing season: March–October, early growing season: May–July, and late growing season: August–October) for each year relevant to seasonal hydrology and plant phenology. May through July represents mid-peak growing season and a time period of high to low river discharge, generally freshwater conditions, and early tropical cyclone activity. August through October represented late growing season, low river discharge, and higher salinities, and late tropical cyclone activity. Including data from colder months (January–April and November–December) in the analyses generally lowered the strength of significant relationships between abiotic conditions and vegetation response.

Marsh inundation and salinity metrics were also calculated with and without tropical cyclone impacts. Cyclone impacts were removed by first compiling information on all Gulf of Mexico cyclones over the 2007 through 2021 time period using archived tropical cyclone reports (National Hurricane Center, NOAA) then identifying corresponding anomalous patterns of water level and salinity at each Coastwide Reference Monitoring System station. Datasets were partitioned into those with and without storm-related water levels and salinity.

To test differences in hydrology and salinity between areas that experienced dieback and those where dieback did not occur, we focused our analysis on dieback and non-dieback Coastwide Reference Monitoring System stations where *Phragmites* cover was greater than 20% for ≥1 years of the 16-year time series (*n* = 36). Stations were classified as freshwater and intermediate salinity marshes. We tested differences between dieback and non-dieback marshes in mean daily marsh inundation, mean maximum marsh inundation, mean daily maximum salinity, and mean maximum salinity during the early growing season May through July, a time period where abiotic variables were most strongly related to vegetation response.

## Vegetation state conceptual framework for testing empirical relationships

To test relationships between vegetation cover and marsh inundation and salinity, vegetation cover was classified into one of six vegetation states: Peak, Decline, Dieback, Impact, Recovery, and Resumption that reflect the relative cover and the trajectory from previous year's (Fig. 6). Except for Peak coverage, the classification of the other states was dependent on the prior year's state. Peak cover was classified as the highest 90th percentile of vegetation coverage at each site across the 16-year time series. Declines in cover were considered lower magnitude reduction in cover from the previous year in years other than the dieback years (2012 and 2013). Dieback in contrast was a high magnitude loss of cover in 2012 and 2013. Impact periods were a following a decline or dieback. Recovery was an increase in cover following a decline, dieback, or impact state. Resumption was a stasis following a recovery but lower cover than during Peak states. This framework was used to test differences in marsh inundation and salinity (daily mean and maximum) over the growing season, March through October among states using analysis of variance (ANOVA) and test relationships between vegetation cover and environmental variables within states using regression analysis. For these analyses, marsh flooding and salinity dynamics over the entire growing season were included to capture the strong seasonal variation and relative influence of freshwater river discharge in the spring and higher salinities associated with lower discharge, coastal set-up, and tropical storms in the late summer, early fall (Supplementary Fig. 8).

## Relationship between marsh inundation and salinity and *Phragmites* cover

Multiple linear regression was used to examine the influence of marsh inundation and salinity on the percent cover of *Phragmites* during specified time periods (i.e., May–October, May–July, and August–October) and the population states of *Phragmites*. Additionally, we used a Welch's ANOVA (unequal variances) to test differences among states in marsh inundation and salinity metrics[87]. The Games-Howell test was used for all multiple pairwise comparisons.

To test the potential for thresholds, we conducted sigmoidal regression analyses[88,89] that included relative percent cover as the dependent variable and salinity and marsh inundation during important seasonal time periods as the independent variables. Relative percent cover was used instead of absolute percent cover to account for differences in initial coverages of *Phragmites* across sites. For the sigmoidal analyses, we used the inflection points of the sigmoidal equations to identify thresholds in salinity and inundation for *Phragmites* cover. Sigmoidal regression, exponential limited growth

regression, and Spearman rank correlation analyses were conducted in Sigma Plot Version 13.0 (Systat Software, San Jose, California, USA). Linear regression was used to examine the relationship between vegetative cover, marsh inundation, and salinity across sites and time. An Analysis of Covariance was used to test the decline in vegetation cover over time and the influence of marsh inundation. Data associated with the dieback were removed for this analysis. All statistical analyses were conducted using JMP SAS Pro V.16 (SAS Institute, Cary, North Carolina).

## Reporting summary

Further information on research design is available in the Nature Portfolio Reporting Summary linked to this article.

## Data availability

Source data are provided as a Source Data file. Annual summary data are available as a Supplementary Data 1 file and all raw data are publicly available from the Coastal Protection and Restoration Authority (CPRA) of Louisiana, via the Coastwide Reference Monitoring System program and can be retrieved from the Coastal Information Management System (CIMS) database (http://cims.coastal.louisiana.gov). Source data are provided with this paper.

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

## Acknowledgements

This work was funded by the U.S. Department of Agriculture Cooperative Agreement (Award number: AP20PPQS&T00C189 to Rodrigo Diaz and Tracy Quirk). Fieldwork was conducted and data was provided through the Coastwide Reference Monitoring System. We are grateful for the support and discussions from J. Andy Nyman (Louisiana State University), Leigh Anne Sharpe (Coastal Protection and Restoration Authority), and Barret Fortier (US Fish and Wildlife Service). The manuscript was greatly improved by the insightful and excellent recommendations of Dr. Matthew Kirwan and one anonymous reviewer.

## Author contributions

T.Q. conceived and organized the study, analyzed the data, and created the figures; R.D. received the funding; L.W. contributed figures; T.Q., A.L., M.D.M., R.D., J.T.C., L.W., H.H., and D.J. wrote the paper.

## Competing interests

The authors declare no competing interests.
