## [Peer Review File · Nature Communications]

Reviewers' comments:

Reviewer #1 (Remarks to the Author):

I have provided comments on specific sections of the ms in an attached file.

Overall, I find this an interesting examination of available data for areas subject to a *Phragmites* die off and partial recovery. The authors have done a good job parsing out available data to identify seasonal effects and to isolate the effects of tropical cyclones on salinities and water levels. I think the 'population states' approach is a useful one, although I felt its application could have been described more clearly.

One of the most confounding things about how the work is presented is the lack of a focused introduction to the study area subject to the die-off and its context both in terms of river delta geomorphology and broader coastal Louisiana wetlands. Knowing the area, I was perhaps more sensitive to minor differences in descriptors, e.g., modern vs active, but given the attention that the Mississippi Delta Plain receives in the literature I think terming this part of the delta as MRD will be misleading to many. I suggest the authors select their study site based on the CRMS sites they used (there is really no need to identify one as in Pontchartrain – that geographic description is pertinent only to local audiences), and find a geographic term that is more focused. My suggestion is Birdsfoot Delta but there may be others.

That this study is focused on one species and a relatively small part of the Mississippi River Delta Plain I think it is important that when the authors want to introduce their study in the context of either others areas of *Phragmites* or coastal Louisiana wetlands more generally (they are not all of deltaic origin) that the distinction is clear to the reader.

This leads to a larger issue of how the paper is framed. This is about *Phragmites* – I think that needs to be in the title of the paper. The continental scale drought leads to low water levels in the river – this is the proximal cause rather than the drought. Framing the paper in terms of inundation and the factors that influence it in river dominated systems, might make it more widely applicable, e.g., dams on river or upstream extraction can reduce delta water levels leading to this kind of issue. Relative sea-level rise is also only addressed tangentially. There is no doubt that this area of the coast is subject to high subsidence rates (some more recent specific references should have been included), but tracking surface elevation change in terms of NAVD88 requires a lot of detailed assumptions which are glossed over in the methods section.

An additional context issue which I think needs more attention is the role of the insect scale – this is somewhat dismissed in the introduction, but I was surprised not to see its role discussed in relation to the inundation and salinity stress. There is also within the period of record of the analysis, the Deepwater Horizon oil spill which had impacts in the general vicinity. The authors need to acknowledge this and to the extent possible indicate that it is not a precursor event.

I have several comments on the ms in relation to organization in addition to the context setting points made here. I think this material might be better presented as results followed by discussion and some conclusion, if the journal allows that.

Reviewer #2 (Remarks to the Author):

Summary comments:

Elsley-Quirk et al. describe a large die-off of *Phragmites australis* on the lower Mississippi River Delta and argue that drought-induced increases in salinity, coupled with chronic flooding stress, are responsible for the dieoff. Dieback events both in Louisiana and in other regions of the country have previously been attributed to drought, although this manuscript does it in a more thorough way, and with the benefit of long-term data before and after the dieback event. Though the link to drought was compelling, I am not convinced that the data supports the authors conclusion that chronic sea level rise contributed substantially to the dieback. Additionally, more work is needed to make the manuscript appeal to a broad readership. In particular, the authors should present the work in the context of how coastal ecosystems generally respond to episodic and chronic disturbances, and discuss how the work advances our understanding of other large marsh die-off events that have previously been studied.

Strengths:

The authors provide a thorough discussion of a very large marsh die-back event, and present a convincing argument that it was driven primarily by drought-induced changes in salinity. The manuscript and associated data include a nice, nuanced discussion of timing of abnormal water levels and salinities relative to typical seasonal cycles and periods of plant vulnerability. This is refreshing given the tendency for others to generalize these stressors in terms of annual averages. I also appreciated the conceptual framework (Figure 9a) that divides the dieback and subsequent recovery into various stages. This framework provides the basis for the most important of the statistical analyses. In general, the methods and statistical approaches seem appropriate, and leverage a wonderful long-term dataset.

Weaknesses:

1. The title, abstract, and conclusions all indicate that flooding associated with chronic sea level rise is a substantial driver of the *Phragmites* dieback. I assume this conclusion comes from Fig. 9c which shows that during periods of *phragmites* decline, percent cover decreases with maximum marsh inundation depth during the early growing season. However, other data seems inconsistent with this conclusion. For example, 9b shows that percent cover is lowest when mean inundation is lowest. Figure 10 shows that declining and recovery phases occur for the same mean marsh inundation depth (Fig 10b) and percent time flooded (Fig 10c). Figure 3 suggests that marsh inundation during Tropical Storm Debbie and Hurricane Isaac may have preceded the dieback event, but this is, of course, different than the influence of chronic sea level rise. So in summary, the relationships based on mean marsh inundation data show no relationship or a positive relationship with *phragmites* cover, while only the maximum marsh inundation data seems to support the authors conclusion. One would expect that if chronic sea level rise were to be a substantial factor that it would be expressed in mean (rather than maximum) marsh inundation data.

2. The manuscript is entirely framed around the fate of the lower Mississippi River Delta. While this is an incredibly important area, the manuscript misses opportunities to put the work in the context of broader ecological theory (e.g. responses to press vs. pulse stressors), previous work on this dieback event (Cronin et al., 2020), and other dieback events located elsewhere. For example, there have been other large dieback events in coastal Louisiana (McKee et al., 2004; Silliman et al., 2005), and in other regions of the country (Alber et al., 2008; Hughes et al., 2012). The only reference to these papers is to argue that drivers of dieback are complex and largely unexplained. In reality, all of these papers attribute dieback at least in part to drought. So the authors must show how their work advances our previous understanding.

3. The implications for river management (last paragraph of the manuscript) depend on on establishing a link between chronic sea level rise and phragmites dieback, but there appear to be some contradictions (or at least places where I don't follow the logic). For example, Line 505 states that upstream Mississippi River diversions will lead to reduced freshwater discharge and sediment fluxes for the lower delta. Given the authors conclusion that salinity and chronic flooding are drivers of the dieback, it seems like the authors would suggest that diversions will increase the likelihood of dieback. However, the concluding sentence of the manuscript (509-511) argues the exact opposite: that river management will ENHANCE resiliency and FACILITATE recoveries. Furthermore, Fig 10 seems to show that recovery is not determined by inundation (i.e. periods of decline and recovery occur for similar marsh inundation levels).

4. The manuscript is quite long, and overly descriptive (i.e. much more results than discussion) in some places. Its possible that I missed a more definitive link between inundation and dieback just because of its length. There is just a lot of data. Some example places to shorten or remove include: Lines 129-151 (natural seasonal variation not directly related to the dieback), Figure 7 (examples to illustrate a method used in subsequent figure), Lines 321-331 (discussion of sulfides even though sulfide wasn't measured), Lines 333-344 (only references a supplemental figure), and lines 483-493 (states that drought was severe which is redundant with the intro).

Other comments:

5. Seems like a part of the story that is missing is the expanded footprint of other vegetation species into a niche that was previously occupied by phragmites. Figure 3a shows that total vegetation cover has completely recovered to pre-dieback levels. This implies that the demise of phragmites has led to a long-term increase in biodiversity. Yet, other species get a single sentence in the manuscript (lines 459-462).

6. Figure 3 offers at least some indication that there have been other dieback events of Phragmites during the period of record. Its hard to tell, but maybe one around 2009 and another around 2020. Does the severity or recovery from these events influence the interpretation that chronic flooding is a substantial factor?

7. The data availability statement refers readers to the raw data contained in the CRMS database. It would be helpful to also include the summary data that would be necessary to reconstruct the most important figures.

Reviewer #1 (Remarks to the Author):

I have provided comments on specific sections of the ms in an attached file.

Overall, I find this an interesting examination of available data for areas subject to a *Phragmites* die off and partial recovery. The authors have done a good job parsing out available data to identify seasonal effects and to isolate the effects of tropical cyclones on salinities and water levels. I think the 'population states' approach is a useful one, although I felt its application could have been described more clearly.

Thank you. We have expanded the description and clarified the usefulness of the 'population states' framework in both the Results and Methods sections of the revised manuscript. By classifying each annual vegetation cover as one of six vegetation states (i.e., peak, dieback, decline, impact, recovery, and resumption) based on previous year's trajectories, we parse out environmental conditions associated with each 'state' and test relationships between environmental conditions and vegetation cover with the variability associated with biological time lags removed (e.g., prolonged impact when conditions are favorable; lag time in regeneration and recolonization). We added detail to both the Results section (L. 264 – 288) and the Methods section (L. 671 – 689).

One of the most confounding things about how the work is presented is the lack of a focused introduction to the study area subject to the die-off and its context both in terms of river delta geomorphology and broader coastal Louisiana wetlands. Knowing the area, I was perhaps more sensitive to minor differences in descriptors, e.g., modern vs active, but given the attention that the Mississippi Delta Plain receives in the literature I think terming this part of the delta as MRD will be misleading to many. I suggest the authors select their study site based on the CRMS sites they used (there is really no need to identify one as in Pontchartrain – that geographic description is pertinent only to local audiences), and find a geographic term that is more focused. My suggestion is Birdsfoot Delta but there may be others.

We have revised the manuscript to address the recommendation for greater geographic and geomorphic context. We describe the study region more specifically and distinguished the birdsfoot delta from the greater Mississippi River Delta plain. We have also added more detail about the geographic scale of investigation and geologic context. For example, while the dieback was initially reported in the birdfoot delta (BFD), we tested CRMS vegetation cover data for evidence of dieback at multiple spatial scales: 1) at all CRMS stations in the BFD (n = 14), 2) at all CRMS stations across the coast where *Phragmites* was present (n = 74), 3) at all CRMS sites where *Phragmites* comprised > 20% cover for at least one year (n = 33; Fig. 2a), and 4) at a random subset of saline, brackish and freshwater marshes without *Phragmites* present (n = 30). The results showed that dieback between 2012 and 2017 was concentrated in the BFD but also occurred at two sites outside of the BFD – all where *Phragmites* comprised > 20% cover for at least one year. However, 20 sites farther west and in the Chenier Plain region where *Phragmites* was dominant did not experience dieback. The comparison between dieback and non-dieback sites provides important insights into the conditions (i.e., higher inundation) that increased the vulnerability to the dieback caused by drought-induced salinity incursion.

That this study is focused on one species and a relatively small part of the Mississippi River Delta Plain I think it is important that when the authors want to introduce their study in the context of either others

areas of *Phragmites* or coastal Louisiana wetlands more generally (they are not all of deltaic origin) that the distinction is clear to the reader.

We have expanded our analysis to include an examination of all plant species and other locations. The updated analysis and findings further support the theme of the study, which is that the impacts of acute disturbances are more severe and the degree of recovery is lower in areas undergoing chronic stress. To fully address the reviewer's suggestions, we added discussion of the measured spatial variability in chronic stress that leads to a lower recovery and a longer-term decline in vegetation cover, specifically why sites in the Birdfoot delta were most impacted, had lower recovery, and are experiencing longer-term decline.

This leads to a larger issue of how the paper is framed. This is about *Phragmites* – I think that needs to be in the title of the paper.

Our subsequent analysis shows that the dieback affected multiple plant species in *Phragmites*-dominated marshes. Given this finding, we think it is appropriate to use “vegetation dieback” in the title. However, we included detail about the dominance of *Phragmites* in the communities affected in the paper and discussed possible reasons why dieback events typically target a single species and why dieback occurred in *Phragmites*-dominated communities and had a large impact on *Phragmites* in this study.

The continental scale drought leads to low water levels in the river – this is the proximal cause rather than the drought. Framing the paper in terms of inundation and the factors that influence it in river dominated systems, might make it more widely applicable, e.g., dams on river or upstream extraction can reduce delta water levels leading to this kind of issue.

We added a discussion paragraph to address the reviewer's suggestion that while drought was the external driver of the dieback in this case, other factors that lower river water levels can result in a similar outcome. We note that the proximate cause was not low river water but the salt-water incursion that occurred in the region. We discuss how drought was a driver for this particular dieback and include citations for several papers that also relate drought to vegetation dieback.

Relative sea-level rise is also only addressed tangentially.

We added new statistical analyses and results (ANCOVA & Fig. 8), and text that describe a direct link between vegetation cover decline and marsh inundation increase over the growing season.

There is no doubt that this area of the coast is subject to high subsidence rates (some more recent specific references should have been included), but tracking surface elevation change in terms of NAVD88 requires a lot of detailed assumptions which are glossed over in the methods section.

As the reviewer suggested, we added much more detail on our calculations of marsh inundation and surface elevation change in the revised Results and Methods sections. Elevation change was measured using biannual surface elevation table readings which were referenced to NAVD88. This methodology is widely recognized as the most accurate method to track recent marsh elevation changes. We also added recent and relevant citations for the elevation change and subsidence rates.

An additional context issue which I think needs more attention is the role of the insect scale – this is somewhat dismissed in the introduction, but I was surprised not to see its role discussed in relation to the inundation and salinity stress.

There is also within the period of record of the analysis, the Deepwater Horizon oil spill which had impacts in the general vicinity. The authors need to acknowledge this and to the extent possible indicate that it is not a precursor event.

Though our findings illustrate the importance of drought-related salinity intrusion and underlying flood stress to vegetation dieback, there are, as the reviewer suggests, many stressors in this system. We added a paragraph to the Discussion section (L 406 – 434) that addresses the (potential) effects of multiple factors including the 2010 Deepwater Horizon oil spill, high nutrient concentrations in the Mississippi River, and the scale infestation.

I have several comments on the ms in relation to organization in addition to the context setting points made here. I think this material might be better presented as results followed by discussion and some conclusion, if the journal allows that.

We have addressed all of the reviewer's comments and suggestions on the manuscript in the revised version and, as suggested by the reviewer, we reformatted the manuscript to separate the Results and Discussion sections.

Reviewer #2 (Remarks to the Author):

Summary comments:

Elsy-Quirk et al. describe a large die-off of *Phragmites australis* on the lower Mississippi River Delta and argue that drought-induced increases in salinity, coupled with chronic flooding stress, are responsible for the dieoff. Dieback events both in Louisiana and in other regions of the country have previously been attributed to drought, although this manuscript does it in a more thorough way, and with the benefit of long-term data before and after the dieback event. Though the link to drought was compelling, I am not convinced that the data supports the authors conclusion that chronic sea level rise contributed substantially to the dieback.

We hope that the revised manuscript will convince the reviewer and other readers that 1) chronic flooding increased the susceptibility to the acute dieback caused by drought-induced salinity incursion and 2) low elevations and an increase in flooding over time is directly related to both the interannual and longer-term loss of vegetation cover in the birdfoot delta. We illustrate the former point with a revised Result description and Figure 4 and the latter point with a significant ANCOVA describing the decline in vegetation cover over time related to an increase in marsh inundation over the growing season, Fig 5 c,d and new Figure 8 that shows vegetation cover decline with higher flood depths. The vegetation cover data (new Fig. 3) also illustrates a loss of species richness and an increase in more flood-tolerant species over the long-term. We added additional discussion to clarify the role of SLR and subsidence in influencing the increase in marsh inundation adjacent to the Mississippi River in the Birdsfoot delta as compared to other sites that tended to be less impacted by the drought.

Additionally, more work is needed to make the manuscript appeal to a broad readership. In particular, the authors should present the work in the context of how coastal ecosystems generally respond to episodic and chronic disturbances, and discuss how the work advances our understanding of other large marsh die-off events that have previously been studied.

As suggested by the reviewer, we reframed the manuscript to appeal to a broader readership and included more text dedicated to how coastal ecosystems respond to episodic and chronic disturbances. We also highlighted how our study advances our understanding of marsh dieback events by quantifying the simultaneous effects of chronic flood stress and an acute drought-related disturbance on vegetation dieback. Using continuous field data, we were able to identify tipping points for vegetation dieback and conditions associated with recovery and more gradual longer-term decline, which advance our understanding of the role of chronic stress on the impact and recovery from episodic disturbances. The findings have clear management implications pointing to the importance of mitigating the underlying stresses to foster resilience to both acute and persistent causes of vegetation loss.

Strengths:

The authors provide a thorough discussion of a very large marsh die-back event, and present a convincing argument that it was driven primarily by drought-induced changes in salinity. The manuscript and associated data include a nice, nuanced discussion of timing of abnormal water levels and salinities relative to typical seasonal cycles and periods of plant vulnerability. This is refreshing given the tendency for others to generalize these stressors in terms of annual averages. I also appreciated the conceptual framework (Figure 9a) that divides the dieback and subsequent recovery into various stages. This framework provides the basis for the most important of the statistical analyses. In general, the methods and statistical approaches seem appropriate, and leverage a wonderful long-term dataset.

Weaknesses:

1. The title, abstract, and conclusions all indicate that flooding associated with chronic sea level rise is a substantial driver of the Phragmites dieback. I assume this conclusion comes from Fig. 9c which shows that during periods of phragmites decline, percent cover decreases with maximum marsh inundation depth during the early growing season. However, other data seems inconsistent with this conclusion. For example, 9b shows that percent cover is lowest when mean inundation is lowest. Figure 10 shows that declining and recovery phases occur for the same mean marsh inundation depth (Fig 10b) and percent time flooded (Fig 10c). Figure 3 suggests that marsh inundation during Tropical Storm Debbie and Hurricane Isaac may have preceded the dieback event, but this is, of course, different than the influence of chronic sea level rise. So in summary, the relationships based on mean marsh inundation data show no relationship or a positive relationship with phragmites cover, while only the maximum marsh inundation data seems to support the authors conclusion. One would expect that if chronic sea level rise were to be a substantial factor that it would be expressed in mean (rather than maximum) marsh inundation data.

We have thoroughly addressed the Reviewer's comments in the revised manuscript (described also above) by adding new statistical analysis, Figure 8, and clearer interpretation of the data and results. To address the specific points of the Reviewer:

1) Fig. 9 b (in the original MS) only focuses on the conditions during Peak and Dieback (not annual Declines). Dieback in was associated with relatively low water during a drought and high salinity. Annual Declines shown in Fig. 9 c increased in magnitude when early season maximum marsh inundation depth increased. This early season effect of maximum water level adjacent to the Mississippi River in the Birdfoot delta (BFD) was associated with major river discharge events. So yes, this is not a RSLR effect but a decline in the vegetation during the summer immediately following high spring/early summer flooding. The relationship was also significant albeit less strong when inundation was averaged over the entire growing season. For the RSLR effect, we added analyses and figures that show the significance of mean inundation over the entire growing season on the temporal decline in vegetation cover in the (BFD; ANCOVA and Fig. 8). We discuss the relevance of both influences (early summer extreme flood events and longer term increase in marsh inundation) on vegetation declines.

2) Figure 10 shows indeed that mean flooding over the growing season was similar during annual Declines and Recoveries. Annual Declines were more sensitive to early growing season flooding. If we were to plot the population states with respect to May – Jul flooding, there would be a large difference between Declines and Recoveries. This is illustrated by comparing Fig. 7 b and c. From Fig. 7c, you can see that Recoveries only occur at optimal levels of early season flood depths (around 20 cm). At these flood depths on Fig. 7b, declines are pretty low and increase dramatically when flood depths average 4 – 90 cm May – July. For the population state figures we decided to show salinity and flooding data averaged over the entire growing season to capture the average conditions, which incorporates large seasonality in water levels and salinities (Supplemental Fig. 8) and reduces the importance of infrequent events such as high river discharge early in the growing season and cyclones later in the growing season. We improved our explanation and implications of the time scale differences in the revised manuscript.

3) Tropical Storm Debbie preceded the dieback but Hurricane Isaac immediately followed the initial measurements of dieback in summer 2012 (Supplemental Fig. 3). We clarified their role in exacerbating the effects of the drought by causing very high salinities and flooding over a relatively short time period. We agree that the effects of storm flooding are different from RSLR.

2. The manuscript is entirely framed around the fate of the lower Mississippi River Delta. While this is an incredibly important area, the manuscript misses opportunities to put the work in the context of broader ecological theory (e.g. responses to press vs. pulse stressors), previous work on this dieback event (Cronin et al., 2020), and other dieback events located elsewhere. For example, there have been other large dieback events in coastal Louisiana (McKee et al., 2004; Silliman et al., 2005), and in other regions of the country (Alber et al., 2008; Hughes et al., 2012). The only reference to these papers is to argue that drivers of dieback are complex and largely unexplained. In reality, all of these papers attribute dieback at least in part to drought. So the authors must show how their work advances our previous understanding.

We revised this manuscript to address the Reviewer's suggestions to put our research in the context of broader ecological theory by incorporating recent and relevant literature from global analyses to other ecosystems. We also expanded the discussion of other dieback events and previous work on this dieback in both the Introduction and Discussion sections. This allowed us to illustrate how our work specifically

advances our understanding of how underlying stresses increase the vulnerability to and can limit the recovery from acute disturbance events.

3. The implications for river management (last paragraph of the manuscript) depend on establishing a link between chronic sea level rise and phragmites dieback, but there appear to be some contradictions (or at least places where I don't follow the logic). For example, Line 505 states that upstream Mississippi River diversions will lead to reduced freshwater discharge and sediment fluxes for the lower delta. Given the authors conclusion that salinity and chronic flooding are drivers of the dieback, it seems like the authors would suggest that diversions will increase the likelihood of dieback. However, the concluding sentence of the manuscript (509-511) argues the exact opposite: that river management will ENHANCE resiliency and FACILITATE recoveries. Furthermore, Fig 10 seems to show that recovery is not determined by inundation (i.e. periods of decline and recovery occur for similar marsh inundation levels).

We have revised and clarified this section of the Discussion. The implications of this study are that in order to increase the resiliency of the marsh vegetation particularly in the BFD, flooding needs to be reduced, specifically marsh elevations need to be raised. This could be accomplished through direct thin-layer deposition from navigation dredging and/or through strategic crevasse placement that would increase the sediment input to these areas.

4. The manuscript is quite long, and overly descriptive (i.e. much more results than discussion) in some places. It's possible that I missed a more definitive link between inundation and dieback just because of its length. There is just a lot of data. Some example places to shorten or remove include: Lines 129-151 (natural seasonal variation not directly related to the dieback), Figure 7 (examples to illustrate a method used in subsequent figure), Lines 321-331 (discussion of sulfides even though sulfide wasn't measured), Lines 333-344 (only references a supplemental figure), and lines 483-493 (states that drought was severe which is redundant with the intro).

We have reduced the length of the manuscript overall (by 2 pages of non-methods text). The sections highlighted by the reviewer were removed.

Other comments:

5. Seems like a part of the story that is missing is the expanded footprint of other vegetation species into a niche that was previously occupied by phragmites. Figure 3a shows that total vegetation cover has completely recovered to pre-dieback levels. This implies that the demise of phragmites has led to a long-term increase in biodiversity. Yet, other species get a single sentence in the manuscript (lines 459-462).

This is an excellent point and we have updated the analysis and discussion to include the effect of the dieback on other species as well as the changes in species richness and plant composition during and after the dieback.

6. Figure 3 offers at least some indication that there have been other dieback events of Phragmites during the period of record. It's hard to tell, but maybe one around 2009 and another around 2020. Does

the severity or recovery from these events influence the interpretation that chronic flooding is a substantial factor?

Because these declines in cover were not total or severe vegetation loss, these were classified as “declines” in cover, according to the defined Vegetation States. The magnitude of the declines in cover were significantly related to maximum marsh inundation (previous Fig 9 c). This indicates that maximum marsh inundation during the early growing season (May – Jul) is important in causing a reduction in *Phragmites*.

7. The data availability statement refers readers to the raw data contained in the CRMS database. It would be helpful to also include the summary data that would be necessary to reconstruct the most important figures.

We agree. We have added a Supplementary Data Table that includes annual summary data for vegetation percent cover, early growing season (May – Jul) non-storm mean marsh inundation and mean daily maximum salinity, Jun – Jul mean maximum marsh inundation and salinity, Aug – Oct mean maximum inundation and salinity, and full growing season (May – Oct) mean marsh inundation, percent time flooded and mean daily maximum salinity.

REVIEWERS' COMMENTS

Reviewer #1 (Remarks to the Author):

The authors have done an excellent job addressing my comments on an earlier version of the paper. I think this is now much clearer and represents the system more appropriately. I have a few rather minor points:

Lines 137-138 presents a rather definitive statement about RSLR rates and what marshes can tolerate. I suggest this be moderated a bit with at least a 'may exceed' as this is just one of a number of studies and local measurements of surface change or accretion can exceed 1.3cm/yr.

Line 185. As in my previous comments the distinction between basins such as Terrebonne and Pontchartrain is not necessary and somewhat meaningless the way the paper has been rewritten.

Line 189 - similarly you haven't distinguishes between Chenier and Delta Plain I don't think. Suggest just southeast and southwest Louisiana would do.

Line 208. May through July is not the major period of 'storm flooding' in Louisiana. Peak of hurricane season is Sept. What flooding are you referring to?

Figure 4. The graphs should use dieback not dieoff. Also the caption refers to regions - I think these are CRMS sites not parts of the coast?

Line 234 - I think the National Weather Service refers to this as Debby not Debbie.

Line 245 - see previous comments on Pontchartrain as a label.

Reviewer #2 (Remarks to the Author):

Elsy-Quirk et al. describe a large die-off of *Phragmites australis* on the lower Mississippi Delta and present a convincing argument that it was driven primarily by drought-induced changes in salinity. Although there have been other papers about drought-induced dieback in the Southeastern U.S., this manuscript is unique in that it leverages a wonderful long-term dataset that includes vegetation data before the initiation of dieback. This allows the authors to quantify specific environmental conditions that are responsible for dieback in ways that have not previously been possible. Figure 7 is a prime example of the quantitative analysis that is unparalleled in other papers on the topic.

In my first review, I identified 4 significant weaknesses that have now all been addressed. First and most importantly, the authors added analyses related to the role of inundation. I found the new Figures 7 and 8 to be particularly convincing. Second, the authors have thoroughly revised the introduction to present other cases of vegetation dieback, with examples linked to both inundation and drought. Third, the

implications to river management have been revised to focus on dredge spoil placement rather than diversions. I'm not sure the revised paragraph is particularly useful, but it is at least accurate. Fourth, The manuscript has been significantly shortened by removing less relevant text.

As before, the manuscript and associated data include a nice, nuanced discussion of timing of abnormal water levels and salinities that is refreshing given the tendency for others to generalize these stressors in terms of annual averages. I also appreciated the conceptual framework (now Figure 6) that divides the dieback and subsequent recovery into various stages. This framework provides the basis for the most important of the statistical analyses. In general, the methods and statistical approaches seem appropriate, and leverage a wonderful long-term dataset.

I rarely see papers that improve this much in revision, and I now thoroughly recommend its publication in nature communications.

A couple very minor comments:

49- desert not dessert

152- very long sentence

484- In this paragraph, consider Coleman and Kirwan, 2018 Earth Surface Processes and Landforms as an additional citation for the effects of dieback on sediment erosion and accretion.

This is up to the authors, but I would consider ending the manuscript with the current drought conditions, combined with an abridged version of the impacts of dieback (paragraph beginning on line 484). There was an excellent sentence earlier in the discussion that read "As we write this paper, drought conditions have caused two consecutive years of historically low water levels in the Mississippi River." If it were me, I'd make that the lead sentence of the final paragraph. It gives the paper more sense of timeliness and could replace a somewhat generic implications section.

REVIEWERS' COMMENTS

Reviewer #1 (Remarks to the Author):

The authors have done an excellent job addressing my comments on an earlier version of the paper. I think this is now much clearer and represents the system more appropriately. I have a few rather minor points:

Lines 137-138 presents a rather definitive statement about RSLR rates and what marshes can tolerate. I suggest this be moderated a bit with at least a 'may exceed' as this is just one of a number of studies and local measurements of surface change or accretion can exceed 1.3cm/yr.

We have added the word “may” to the sentence as the Reviewer suggested.

Line 185. As in my previous comments the distinction between basins such as Terrebonne and Pontchartrain is not necessary and somewhat meaningless the way the paper has been rewritten.

Okay, we deleted the location information in the figure legend.

Line 189 - similarly you haven't distinguishes between Chenier and Delta Plain I don't think. Suggest just southeast and southwest Louisiana would do.

We revised the sentence as suggested by the Reviewer.

Line 208. May through July is not the major period of 'storm flooding' in Louisiana. Peak of hurricane season is Sept. What flooding are you referring to?

Maximum flood depths represent storm flooding events even in May through July. Tropical cyclones and storms also occur May – July. This is in contrast to mean flood depths that would represent average, not extreme, conditions.

Figure 4. The graphs should use dieback not dieoff. Also the caption refers to regions - I think these are CRMS sites not parts of the coast?

We revised the graph and legend as suggested.

Line 234 - I think the National Weather Service refers to this as Debby not Debbie.

Yes, thank you. We revised this misnomer!

Line 245 - see previous comments on Pontchartrain as a label.

We removed the distinction as suggested.

Reviewer #2 (Remarks to the Author):

Elsy-Quirk et al. describe a large die-off of *Phragmites australis* on the lower Mississippi Delta and present a convincing argument that it was driven primarily by drought-induced changes in salinity. Although there have been other papers about drought-induced dieback in the Southeastern U.S., this manuscript is unique in that it leverages a wonderful long-term dataset that includes vegetation data before the initiation of dieback. This allows the authors to quantify specific environmental conditions that are responsible for dieback in ways that have not previously been possible. Figure 7 is a prime example of the quantitative analysis that is unparalleled in other papers on the topic.

In my first review, I identified 4 significant weaknesses that have now all been addressed. First and most importantly, the authors added analyses related to the role of inundation. I found the new Figures 7 and 8 to be particularly convincing. Second, the authors have thoroughly revised the introduction to present other cases of vegetation dieback, with examples linked to both inundation and drought. Third, the implications to river management have been revised to focus on dredge spoil placement rather than diversions. I'm not sure the revised paragraph is particularly useful, but it is at least accurate. Fourth, The manuscript has been significantly shortened by removing less relevant text.

As before, the manuscript and associated data include a nice, nuanced discussion of timing of abnormal water levels and salinities that is refreshing given the tendency for others to generalize these stressors in terms of annual averages. I also appreciated the conceptual framework (now Figure 6) that divides the dieback and subsequent recovery into various stages. This framework provides the basis for the most important of the statistical analyses. In general, the methods and statistical approaches seem appropriate, and leverage a wonderful long-term dataset.

I rarely see papers that improve this much in revision, and I now thoroughly recommend its publication in nature communications.

A couple very minor comments:

49- desert not dessert

Ha ha! Yes, we fixed the spelling error!

152- very long sentence

We divided the sentence into two shorter sentences.

484- In this paragraph, consider Coleman and Kirwan, 2018 Earth Surface Processes and Landforms as an additional citation for the effects of dieback on sediment erosion and accretion.

Thank you for the reference. We added it to the sentence.

This is up to the authors, but I would consider ending the manuscript with the current drought conditions, combined with an an abridged version of the impacts of dieback (paragraph beginning on line 484). There was an excellent sentence earlier in the discussion that read "As we write this paper, drought conditions have caused two consecutive years of historically low water levels in the Mississippi River." If it were me, I'd make that the lead sentence of the final paragraph. It gives the paper more sense of timeliness and could replace a somewhat generic implications section.

We agree. We revised the Implication section by beginning with the suggested statement and incorporating an abridged version of the last discussion paragraph.